# The Anti-Müllerian Hormone as Endocrine and Molecular Marker Associated with Reproductive Performance in Holstein Dairy Cows Exposed to Heat Stress

**DOI:** 10.3390/ani14020213

**Published:** 2024-01-09

**Authors:** Luis A. Contreras-Méndez, Juan F. Medrano, Milton G. Thomas, R. Mark Enns, Scott E. Speidel, Guillermo Luna-Nevárez, Pedro A. López-Castro, Fernando Rivera-Acuña, Pablo Luna-Nevárez

**Affiliations:** 1Departamento de Ciencias Agronómicas y Veterinarias, Instituto Tecnológico de Sonora, Ciudad Obregón 85000, Mexico; 2Department of Animal Science, University of California, Davis, CA 95616, USA; 3Texas A&M AgriLife Research, Beeville, TX 78102, USA; 4Department of Animal Sciences, Colorado State University, Fort Collins, CO 80523, USA

**Keywords:** cattle, fertility, GWAS, heat stress, polymorphisms, validation

## Abstract

**Simple Summary:**

Reproduction in Holstein cows is a critical factor influencing herd profitability. Heat stress (HS) caused by extremely warm climatic conditions disrupts several physiological processes, lowering fertility. Under the prospect of global warming, the selection of heat-tolerant cows able to maintain adequate reproductive performance during summer is favorable. The measurement of serum Anti-Müllerian hormone (AMH) has been proposed as a biomarker for fertility in Holstein cows, as well as single nucleotide polymorphisms (SNPs) associated with AMH. However, the HS that occurs in the summer of semi-arid regions appears to affect the predictability of the AMH marker(s). It could be due to a reduced ability of the follicle to synthesize AMH under HS conditions. In the current study, serum AMH was proved as an endocrine marker for fertility in cows subjected to a reproductive management program during summer but only if the heat stress did not exceed a moderate threshold. Polymorphisms within the AMH gene were validated as genetic markers associated with reproductive performance in cows exposed to either moderate or severe HS. However, due to the limited amount of variation accounted by these tools, their utility may be in conjunction with a genomic breeding value approach to help improve fertility in Holstein cows exposed to HS.

**Abstract:**

Anti-Müllerian hormone (AMH) is proposed as a biomarker for fertility in cattle, yet this associative relationship appears to be influenced by heat stress (HS). The objective was to test serum AMH and AMH-related single nucleotide polymorphisms (SNPs) as markers potentially predictive of reproductive traits in dairy cows experiencing HS. The study included 300 Holstein cows that were genotyped using BovineSNP50 (54,000 SNP). A genome-wide association study was then executed. Nine intragenic SNPs within the pathways that influence the *AMH* gene were found important with multiple comparisons adjustment tests (*p* < 1.09 × 10^−6^). A further validation study was performed in an independent Holstein cattle population, which was divided into moderate (MH; *n* = 152) and severe heat-stressed (SH; *n* = 128) groups and then subjected to a summer reproductive management program. Serum AMH was confirmed as a predictor of fertility measures (*p* < 0.05) in MH but not in the SH group. Cows were genotyped, which revealed four SNPs as predictive markers for serum AMH (*p* < 0.01), reproductive traits (*p* < 0.01), and additional physiological variables (*p* < 0.05). These SNPs were in the genes *AMH*, *IGFBP1*, *LGR5*, and *TLR4*. In conclusion, serum AMH concentrations and AMH polymorphisms are proposed as predictive markers that can be used in conjunction with genomic breeding value approaches to improve reproductive performance in Holstein cows exposed to summer HS conditions.

## 1. Introduction

Fertility is important for the efficiency of dairy herds, as it accounts for one of the major costs of production. A critical factor affecting fertility in cows managed in semi-arid areas is heat stress (HS), which interferes with reproductive processes and causes a pronounced decline in the conception rate [1]. Due to their accelerated metabolic state, lactating dairy cows are sensitive to HS because heat production surpasses body heat loss, elevating internal temperature [2]. Under HS, the inability to dissipate internal heat load is primarily responsible for impaired reproduction [3].

Several strategies have been reported to study the effects of HS on lactating cow fertility. The immediate strategy to reduce declines in fertility is to properly assess HS and provide a cooling program to dairy cows on the farm, regardless of their physiological stage [4]. Other short- and long-term management solutions to mitigate fertility declines during HS include the use of embryo transfer technology and timed artificial inseminations (TAI) from germplasm identified as thermo-tolerant with genome-assisted breeding values, lowering the rates of disease and mastitis, and feeding care. The best long-term strategy is an effective breeding program to increase fertility and tolerance to HS [5].

In the last century, selection programs in dairy cattle systems were focused primarily on improving milk yield traits [6], which caused a reduction in the genetic merit for fertility [7]. Currently, selection programs include reproductive and health functional traits in addition to those related to milk production [6,8,9]. However, for the dairy industry of the future, general selection for fertility and HS tolerance is crucial [10]. The selection indices should take into account both fertility and HS tolerance in order to identify superior animals. For thermotolerance and fertility, numerous genes and genetic markers have been shown to exist; these can be used in the marker- and genome-assisted selection programs already being used in the global dairy industries [11]. However, to execute a genomic study, it is first necessary to define strategies to identify and monitor appropriate indicators of HS [12]. Indicators of thermo-tolerance studied are body temperature, respiratory rate, and hormones such as cortisol, T3, and T4 [13]. A thermotolerance indicator involving physiological traits (i.e., rectal temperature and respiratory rate) and milk yield was developed to study genes and molecular mechanisms associated with heat stress in dairy cattle [14]. Concomitantly, a basic regression model was employed to create a thermotolerance indicator based on rectal temperature and feed intake data gathered from ewes subjected to HS [15,16].

Evaluations of new measurable traits, such as endocrine or molecular markers for reproductive phenotypes, offer potential new tools to help genetic improvement in fertility in cattle [17]. A phenotypic trait with high variability, heritability, and repeatability that is also associated with fertility could be a good candidate to include in selection strategies [18]. The Anti-Müllerian hormone (AMH) has emerged as a potential biomarker for fertility in dairy cows [19]. Concentrations of AMH in cattle have been reported to have high variability among animals and high repeatability within individual animals, and it is considered a moderately heritable trait (0.36 to 0.46) [20]. Therefore, the diurnal serum AMH concentration has been proposed as a reproductive management tool. The success of this tool has been reported in breeding and superovulation protocols in dairy heifers and lactating cows [21].

Serum AMH concentration is reported to be very stable among age groups (from 1 to 9 years old) and between different stages of the estrus cycle in Holstein cows. However, reference values for AMH concentration differed between *Bos indicus* (770 pg/mL) and *Bos taurus* (330 pg/mL) cattle breeds. A previous report on beef cows detected a decrease in serum AMH during summer (321.4 pg/mL) compared to winter (1018.8 pg/mL), which appeared to be associated with an elevation of the temperature–humidity index [22]. As summer is typically characterized by warm environmental conditions, it provides evidence and needs to identify a threshold value for THI at which serum AMH concentrations are affected. The establishment of this AMH threshold and a test executable on a dairy farm would enable farmers to selectively breed cows with elevated AMH levels in order to increase the reproductive efficiency in their herds [23]. In parallel, genomic technologies make it possible to identify functional genes associated with serum AMH and thermotolerance traits that are directly involved in reproductive processes, as well as to discover and validate specific molecular markers that could play a critical role in enhancing fertility-related physiological events under warm climatic conditions [24].

The efficacy of endocrine and molecular markers as predictors appears to be influenced by the surrounding environment, especially in semi-desert regions where lactating cows often experience HS [22,25]. Heat stress during summer is a major factor responsible for disrupting several reproductive events, and it is defined as an imbalance between the amounts of energy absorbed from various sources, such as the body’s metabolism and environmental conditions, and the energy dissipation system, which leads to an increase in body temperature [26].

Mammalian thermoregulatory mechanisms such as conduction, convection, and radiation are the effective pathways for the removal of body heat. These pathways become insufficient as ambient temperature approaches body temperature, leaving evaporation as the only functional strategy to remove body heat [27]. However, when ambient humidity raises, evaporative routes of thermal regulation, such as increased respiratory rate and sweating, become limited, severely affecting the body’s ability to dissipate heat [28]. A high ambient temperature affects heat loss in lactating dairy cows, leading to an increase in body temperature, which is strongly associated with impaired reproduction [29]. Heat stress affects a number of reproductive processes, including the formation of the Corpus Luteum (CL), ovarian follicular growth, gonadotropin synthesis, steroidogenesis, oocyte competence, embryonic growth, and uterine endometrial responses [30,31].

Cellular and molecular impairments appear to be part of the response of lactating dairy cows to HS. When oocytes are exposed to heat shock during maturation, it hinders their ability to reorganize their microtubules and microfilaments and damages spindle apparatus [32]. Oocytes that were exposed to HS showed an increase in both apoptotic gene expression and production of reactive oxygen species (ROS), which decreased the oocyte’s capacity to split and develop into a blastocyst [33,34]. Taken together, heat stress-induced alterations in oocytes may be associated with fertilization failure, embryo development defects, and consequently low conception rates and (or) high embryo mortality rates [29]. Because of these altered pathways and previous reports involving AMH, our objective was to evaluate associations of serum AMH concentrations and AMH-related single nucleotide polymorphisms (SNPs) with reproductive performance in Holstein dairy cows exposed to heat-stress conditions.

## 2. Materials and Methods

The Institutional Animal Care and Use Committee of the Instituto Tecnologico de Sonora approved all procedures performed on animals (Approval code 2017-0079).

### 2.1. Location and Experimental Population

Three neighboring dairy farms in the Yaqui Valley Sonora, Mexico, collaborated in the current study, and they were managed as intensive milk production systems. The geographic coordinates are 27°21′ North Latitude and 109°54′ West Longitude. The site is located 46 m above sea level. Local climate ranges from dry to semi-humid, with summer rain, an average of 371.6 mm of precipitation annually, and a yearly mean temperature of 23 °C. During the summer, this zone can record ambient temperatures above 48 °C and humidity exceeding 80%, leading to heat stress.

Three hundred lactating and spring-calved Holstein cows with an average body weight of 635.1 ± 32.5 kg, days in milk ranged from 60 to 78, with 2 to 4 lactations, body condition score from 2.5 to 3.5, and 4 to 6 yrs-old were used in this study. Cows received similar management and were kept in shaded stables with free access to a water source and a commercial trace mineral supplement. Floor and shade areas were provided according to the requirements of Holstein cows. A mixed diet was supplied twice daily, which was designed to meet requirements for dairy cows with an average body weight of 650 kg and daily milk production of 30 kg. This ration’s average composition was 3.5% fat and 3.2% crude protein.

### 2.2. Blood Sampling for AMH

Blood samples were collected during middle summer (from 21 July to 31 July) at 17:00 h (i.e., before milking) by puncturing the coccygeal vein from each cow (*n* = 300) and using serum tubes, which were promptly placed on ice. The cows were sampled in a random stage of the estrus cycle, considering that the concentration of AMH is very stable throughout the estrus cycle, allowing us to obtain these data by taking a single measurement at random points in the estrus cycle. After clotting, the samples were centrifuged for 30 min at room temperature (18 to 22 °C). Subsequently, the serum samples were stored at −20 °C until analysis. Serum AMH concentrations were measured using the AMH-bovine specific immunoassay (AMH Fertility Assay^TM^, Minitube of America, Verona, MI, USA), which has a sensitivity of 0.04 pg/mL and an intra-assay coefficient of variation of 2.2%.

### 2.3. SNP Data and Quality Control

An additional blood sample of 3 mL was drawn from each cow (*n* = 300) via venipuncture of the coccygeal vein using disposable sterile syringes. Five drops of the collected whole blood were spotted on Fast Technology for Analysis of Nucleic Acids cards (FTA^®^, Neogen Corporation, Lansing, MI, USA), which were sent to Neogen AgriGenomics Laboratory (Lincoln, NE, USA) for DNA extraction and genotyping. The SNP panel BovineSNP50, which contains 53,218 highly informative SNPs evenly dispersed across the bovine genome, was used to acquire genotypes of each cow.

PLINK v1.07 software was used to implement quality control for the SNPs that were used in genotype-to-phenotype association analyses. Only SNPs that met the following criteria were included in the analyses: a call rate of greater than 95% or a false discovery rate of less than 5%, a missing genotype frequency of less than 5%, a minor allele frequency (MAF) of more than 5%, and no-deviation from the Hardy–Weinberg equilibrium (*p*-value of Chi-square goodness-of-fit test greater than 0.05, χ^2^ > 0.05). The analyses also excluded SNPs with unmapped loci or those on the sex chromosomes. After quality control, 45,832 SNPs were retained for further analyses.

### 2.4. Genome-Wide Association Study (GWAS)

Principal component analysis (PCA) was used to rectify the batch effects and stratification of the test input data. Next, a single-locus mixed model was used for the genomic analysis to investigate the relationships between each SNP marker’s genotype and serum AMH level as a phenotypic observation (i.e., single-marker SNP GWAS). For this purpose, the GWAS was performed SNP by SNP using the software SNP & Variation Suite version 8.8.1 (SVSv8; Golden Helix, Inc., Bozeman, MT, USA, www.goldenhelix.com; accessed on 12 May 2023).

The additive mixed model used was y = Xβ + Za + e, where **y** was the vector of phenotypic observations (AMH); X was the incidence matrix of fixed effects; β was the vector of fixed effects including herd, number of lactations, and the additive effect of the candidate SNP tested for association; Z was the design matrix of random additive genetic effects; a was the vector of random additive genetic effects; and e was the vector of residual effects. A random animal effect was fit in the model to account for population structure. According to this model, a~N (0, Gσ^2^_a_) and e~N (0, Iσ^2^_e_) were assumed, where the additive genetic variance is represented by σ^2^_a_, the residual variance component was σ^2^_e_, the genomic relationship matrix was G, and the identity matrix was I. 

### 2.5. Multiple Testing Adjustment and Linkage Disequilibrium (LD) Analysis

To account for multiple comparisons, the *p*-values derived from the genomic analyses were submitted to the Bonferroni correction test (b = α/n), which assumed independence between SNPs. The number of tests (*n*) was considered the total number of useful SNPs (*n* = 45,832), with the experiment-wise error set at α = 0.05. A nominal *p*-value of 1.09 × 10^−6^ was equivalent to the 5% genome-wide threshold, which corresponded to a value of 5.96 according to the −log10 (*p*-value) scale.

The SNPs that surpassed the Bonferroni adjustment were subjected to a pairwise linkage disequilibrium (LD) test via the ALLELE procedure (SAS software v9.4). The LD was calculated using the R^2^, which is the squared correlation of the alleles at two loci. It determined whether nearby SNP markers were accumulating the effects of distinct or identical genes. In the case a pair of SNPs resulted as highly correlated, then a multi-collinearity test was performed to delineate the linear relationship between SNPs.

### 2.6. Validation Population

An independent cattle population, including Holstein cows (*n* = 280) from two neighboring dairy herds, was selected for the SNP validation. This study was performed to test the candidate SNP markers that the GWAS detected as associated with serum AMH. Cows were spring-calved Holstein, with an average body weight of 651.3 ± 34.7 kg, mean body condition score of 3.0 ± 0.5, and aged 4 to 7. Cows were randomly assigned to one of two experimental groups. The moderate heat-stressed (MH; *n* = 152) group included cows managed during late spring and early summer (from 21 May to 21 July; AT = 28.1 °C, RH = 55.8% and THI = 75.7 units), whereas the severe heat-stressed (SH; n = 128) group included cows managed during middle and late summer (from 21 July to 21 September; AT = 30.6 °C, RH = 71.4% and THI = 82.2 units). The level of HS was established according to the classification described by Collier et al. [35]: (1) absence of HS (<68 units), (2) HS threshold (68–72 units), (3) moderate HS (73–79 units), and (4) severe HS (>79 units).

### 2.7. Reproductive Management

The cows were subjected to an ovulation synchronization protocol that included the insertion of an intravaginal device for delayed progesterone release (CIDR^®^, Zoetis, Parsippany, NJ, USA) at day 0 plus an intramuscular (im) administration of 0.01 mg of GnRH (Fertagyl^®^, Merck & Co., Inc., Rahway, NJ, USA). The CIDR was removed 7 days later and then 25 mg im of prostaglandin F2a (PGF2a, Lutalyse^®^, Zoetis, Parsippany, NJ, USA) was applied. The cows were given a second dose of 0.01 mg of GnRH on day 9, and 24 h later, they were artificially inseminated (FTAI; day 10). Thirty days following FTAI, ultrasound scanning was used to diagnose pregnancy. The number of pregnant cows divided by the total number of FTAI services on day 30 was used to compute the conception rate (CR).

A trans-rectal ultrasonography inspection was performed in cows utilizing a 7.5 MHz transducer (Sonosite MicroMaxx TM^®^, Bothell, WA, USA) to assess the health of their reproductive system, mainly the uterine body and horns. Ovarian activity was also evaluated by measuring dominant follicle diameter (FOL) on day 7 of the synchronization protocol, as well as Corpus Luteum diameter (CL) on day 14 after AI.

### 2.8. Physiological Traits and Climatic Data

During the course of the study, both rectal temperature (RT; °C) and respiratory rate (RR; breaths/min) were measured at 0700 h and 1700 h two times per week. The RT was obtained using a digital thermometer (TES- 1310^®^, Uxcell, Hong Kong, China) with a contact sounding line (type K; 9 cm large), which was inserted into the animal to touch the rectal mucosa, while the RR was determined by visually counting the intercostal movements. Hair coat temperatures (HCT, °C) from the head (HT), shoulder (ST), rump (RT), legs (LT), and udder (UT) were also collected. They were measured with an infrared thermometer (model 568 Fluke^®^, Fluke Corp., Everett, WA, USA; accuracy ± 1°C) positioned approximately 3 m of distance from the animal. A trained graduate student collected these physiological variables during the study to ensure consistency of data collection. A blood sample was collected on day 3 of the study (i.e., the beginning of the follicular wave) to measure the diurnal concentration of serum AMH in both MH (Early spring through early summer) and SH (Middle through late summer) climatic groups.

Data on the ambient temperature (AT; °C) and relative humidity (RH; %) were obtained from the REMAS website (http://www.siafeson.com/remas; accessed on 24 June 2023) by a local meteorological station (Network of Automatic Meteorological Stations of Sonora), which was located near (~500 m) the dairy herds where the study was conducted. Daily records from both AT and RH were used to calculate the temperature–humidity index (THI) via calculations of the following equation: THI = (0.8 AT) + [(RH/100) (AT 14.4)] + 46.4 [36].

### 2.9. SNP Marker Genotyping

An additional blood sample was collected using Vacutainer tubes with EDTA (EDTA-Na2; Venoject^®^, Terumo, Lakewood, CA, USA) by puncturing the jugular vein from each cow at the beginning of the study. The samples were centrifuged at 3500 RMP for 15 min. Then, 200 μL of the leukoplatelet layer was removed by pipet and refrigerated at −20 °C. Next, the DNA was extracted using a commercial kit (DNeasy Blood & Tissue Kits; QIAGEN, Hilden, Germany), following the manufacturer’s instructions. The concentration and purity of the DNA were determined using a NanoDrop automated spectrophotometer (Thermo Fisher Scientific, Waltham, MA, USA).

The ten SNPs described in this study were genotyped using the TaqMan method for allelic discrimination and RT-qPCR (StepOneTM, Applied Biosystems, Foster City, CA, USA) in accordance with the procedures described by Castillo-Salas et al. [37]. The PCR was carried out using the StepOne Real-Time PCR System from Thermo Fisher Scientific (Waltham, MA, USA). Finally, genotype calling and analysis of PCR data were performed using StepOne software (Version 2.3, Life Technologies Corporation, Carlsbad, CA, USA).

### 2.10. Statistics for the Validation Analysis

The mean values of the analyzed traits were calculated using the MEANS and FREQ procedures for continuous and categorical variables, respectively. One-way ANOVA was performed to compare serum AMH, FOL, CL, RT, RR, HT, ST, RT, LT, and UT between MH and SH groups, whereas a Chi-square test was used for comparison of CR between climatic groups. The assumptions of normality and homogeneity of variances were tested via the UNIVARIATE procedure. Allele and genotype frequencies for each polymorphism and deviations from Hardy–Weinberg equilibrium were obtained using the ALLELE program. For the statistical processing of the information, the SAS (Statistical Analysis System) program version 9.4 was used, which included the procedures for genetic studies [38].

Person correlation analysis was performed between serum AMH and ovarian traits (FOL and CL diameters) using PROC CORR within MH and SH climatic groups. Then, a simple linear regression analysis was executed in PROC REG, including serum AMH as a predictor, as well as FOL or CL diameters as response variables. The linear regression equation was *Y*i = β_0_ + β_1_ X, where *Y*i was the response or dependent variable (i.e., FOL or CL diameters), β_0_ was the intercept coefficient, β_1_ was the coefficient of X, and X was the predictor or independent variable (i.e., serum AMH concentration).

A mixed effects statistical model was used to perform an associative study between the genotype and phenotype using the MIXED and GLIMMIX procedures for continuous (AMH, FOL, CL, RT, and RR) and categorical variables (CR), respectively. The associative statistical model was *Y*_ijklm_ = μ + A_i_ + B_j_ + C_k_ + D_l_ + E_m_ + *e*_ijklm_, which included the response variable (*Y*_ijklm_), the fixed effects of the SNP genotype (A_i_), climatic group (B_j_), dairy herd (C_k_) and parity (D_l_), the linear covariate of days in milk (E_m_), and the random effect of sire (*e*_ijklm_). 

If the genotype term was confirmed as a significant source of variation (*p* < 0.05) in the associative statistical analysis then the LSMEANS procedure was performed to make comparisons between average genotypic values, considering the Bonferroni adjustment [25]. The allele substitution effects (i.e., the effect of substituting one allele for another within the population) were determined using a regression model that included the allelic effect as a covariate [39]. Additive genetic effects were obtained using the procedures described by Luna-Nevárez et al. [40]. 

Finally, one-way ANOVA was executed to compare AMH, FOL diameter, and CL diameter values according to the number of favorable SNP genotypes. Later, pairwise comparisons were carried out using Tukey’s HSD test. The Chi-square test was performed to compare the frequencies of CR among favorable SNP genotype groups. A threshold of *p* < 0.05 was established for statistical significance in these analyses.

## 3. Results

### 3.1. Climate and AMH Sampling

During the validation study, moderate HS characterized the climatic conditions in late spring and early summer (i.e., THI ranged from 72 to 79 units), whereas severe HS conditions prevailed in mid- and late summer (i.e., THI ranged from 79 to 84), as observed in Figure 1. Even though RH was highly variable throughout the study, the ambient temperature steadily increased, allowing the THI to reach a value above 79 units. Such climatic conditions suggested a transition from moderate to severe HS in the middle of the summer.

Serum AMH concentrations were first log-transformed to meet the assumptions for a Gaussian distribution. For ease of interpretation, the transformed least-square means results were back transformed. Average values for serum AMH, reproductive traits, and some physiological variables differed (*p* < 0.05) in cows managed during moderate HS relative the cows exposed to severe HS (Table 1). A decrease in serum AMH, FOL, CL, and CR was observed when the THI surpassed the threshold of 79 units, whereas a slight increase was detected in the physiological traits RT and RR. However, hair coat temperatures did not differ between climatic groups. All of these reproductive and physiological variables were normally distributed.

### 3.2. AMH as Endocrine Marker during Summer

In late spring and early summer, strong correlations (r = 0.71 to 0.75, *p* < 0.05) were observed between serum AMH and reproductive traits (i.e., FOL and CL; Table 2). However, this relationship diminished during summer when the THI surpassed 79 units, as correlations dropped (r < 0.30) as per severe HS. A similar associative relationship trend was observed between serum AMH and physiological traits (i.e., RT and RR) throughout the study, although correlations were mostly not significant.

The simple regression analysis detected serum AMH as a predictor for the reproductive traits FOL (R^2^ = 0.56; *p* < 0.01) and CL (R^2^ = 0.52; *p* < 0.01) but only for cows managed under moderate HS (THI = 72 to 79 units; Figure 2A,B). When the THI increased above 79 units, severe HS appeared to affect the ability of serum AMH to predict the ovarian traits FOL and CL (Figure 2C,D).

### 3.3. Genome-Wide Association Study

After applying the quality control criteria, 45,832 SNPs were interpreted as useful and were used in GWAS. Then, the single-locus mixed model identified 81 genome-wide SNPs associated with serum concentrations of AMH (*p* < 0.00001). However, only 20 of these SNPs surpassed multiple testing corrections because they were significant at a lower *p*-value than the Bonferroni adjusted threshold (i.e., 1.09 × 10^−6^; Figure 3). Of these 20 SNPs, 10 were located within genes (i.e., intronic), whereas the remaining 10 SNPs showed an intergenic position, as observed in Table 3.

The Manhattan plot suggested serum AMH concentration in this study was a polygenic trait. These results also suggested that single SNPs only account for a very small percentage of the total additive genetic variance, supporting the statement that the trait “serum AMH concentration” is polygenic. Summary of previous GWAS reports including SNPs and candidate genes associated with serum and plasma AMH concentration are showed in Appendix A.

### 3.4. SNPs Selection

Of the 20 genome-wide SNPs significantly associated with AMH serum levels, 10 intragenic SNPs were selected for use in a validation study because they were located within genes. However, only four intragenic SNPs did not deviate from the Hardy–Weinberg equilibrium (HWE, *X^2^* > 0.05) and met the criteria for minor allele frequency higher than 10% (MAF > 0.10). These four SNPs were rs876084180, rs445674221, rs42849475, and rs8193046, which were in the genes *AMH*, *IGFBP1*, *LGR5*, and *TLR4* (Table 4). Therefore, only these four SNPs were further tested for validation as molecular markers via a genotype-to-phenotype association study.

### 3.5. SNP Markers Validation

The least-square means for AMH and reproductive traits according to SNP genotypes are reported in Table 5. The SNP rs876084180 was associated with serum AMH, FOL, and CL (*p* < 0.01), whereas the SNP rs42849475 was associated with ovarian traits and CR (*p* < 0.05). Interestingly, the SNPs rs445674221 and rs8193046 were predictors of serum AMH (*p* < 0.0001) and all reproductive traits (*p* < 0.01). The SNPs rs445674221 and rs8193046 were also associated with the physiological traits RT and RR (*p* < 0.01), whereas the SNPs rs876084180 and rs42849475 were associated only with RT (*p* < 0.05) as presented in Table 6.

From the SNPs rs876084180, rs445674221, rs42849475, and rs8193046, the most favorable genotypes were AA, GG, GG, and AA, respectively, because they were associated with a better reproductive performance and physiological response. According to the allele substitution effects, the SNP rs876084180 had the highest allele contribution (*p* < 0.0001) on serum AMH levels. However, the SNPs rs445674221 and rs8193046 were the highest contributors to the reproductive traits FOL, CL, and CR (*p* < 0.1). Interestingly, these two SNPs were also the most beneficial contributors for the physiological traits RT and RR (*p* < 0.01), as they were associated with a slight reduction in both variables.

A significant additive fixed effect was confirmed for the SNPs rs876084180, rs445674221, rs42849475, and rs8193046, suggesting that the sum of their individual effects was equal to their combined allele effects.

### 3.6. SNP Effects on Reproductive Phenotypes

An increase in serum AMH and reproductive traits (*p* < 0.05) was observed as the number of favorable genotypes of the four genome-wide significant SNP markers increased from 0 to 4. Serum AMH differed in cows that had 0 to 1, 2 to 3, and 4 favorable genotypes; FOL differed among all favorable genotype categories; CL differed between 0, 1, 2 to 3, and 4 favorable genotypes; and CR differed in cows that had 0, 1 to 3 and 4 favorable genotypes (Figure 4A–D). According to LD analysis, the four significant SNPs were not associated with each other.

## 4. Discussion

Anti-Müllerian hormone (AMH) has been suggested as a potential biomarker associated with ovarian function and fertility traits in dairy cows [41,42]. AMH has been positively associated with the total number of healthy oocytes and follicles in the cow, and this relationship is influenced by climate [22]. An increase in temperature and humidity leads to HS in lactating dairy cows, which affects oocyte quality and development, follicular growth, and embryo viability [43,44]. Severe heat-stressed conditions appear to compromise the predictive ability of the serum AMH concentration as a physiological marker. However, the reliability of AMH as a molecular marker appears to be unaffected by adverse environmental conditions. 

In the current study, we detected that serum AMH appeared to be an endocrine marker for summer reproductive response in Holstein cows, but only when HS did not exceed a moderate threshold. Furthermore, we identified four SNPs associated with serum AMH as molecular markers for reproductive traits, which also were predictors of physiological traits indicative of HS tolerance. These gene variants appeared to influence pathways that regulate serum levels of AMH, such as TGFβ, PI3K/AKT, Wnt, and NF-κB signaling pathways. Perhaps the SNP mutations detected in the current study occur in regulatory regions of the genes *AMH*, *IGFBP1*, *LGR5*, and *TLR4*, which were associated with the variability in circulating concentrations of AMH [23].

In northwestern Mexico, the Yaqui Valley experiences extremely hot weather conditions characterized by an average THI that varies from 75.7 units in late spring to 82.2 units at the end of summer. These THI values indicate heat-stressed climatic conditions progressing from moderate to severe during summer, creating favorable conditions to study the influence of HS on physiological and molecular markers in dairy cattle [25,45]. Therefore, further studies are proposed to elucidate the ability of circulating AMH concentrations to predict fertility in Holstein dairy cows exposed to HS. In addition, the study of AMH polymorphisms as predictors for reproductive performance and thermotolerance appears to be a viable strategy to contribute to the genetic improvement programs of Holstein cows managed under warm environments.

Endocrine and molecular technologies have been performed to identify practical and reliable markers for fertility in cattle. The circulating AMH concentration is proposed as a potential candidate marker due to this trait having high variability, repeatability, and heritability [46,47,48]. In the current study, we first tested serum AMH as an endocrine marker in Holstein cows managed from late spring until the end of the summer. Serum AMH collected at the beginning of a reproductive management program was associated with higher FOL, CL, and CR values and resulted as a predictor for ovarian response (*p* < 0.05) under moderate HS in early summer. However, this predictive ability was not detected in Holstein cows when HS progressed from moderate to severe during late summer. At these later dates, severe HS appeared to affect the ability of the follicle to synthesize AMH, leading to a potential imbalance between follicle size and AMH secretion.

We observed a significant reduction in average serum AMH in Holstein cows after HS increased from moderate to severe (417.26 ± 4.51 to 136.94 ± 4.03 pg/mL). According to Gobikrushanth et al. [23], AMH concentrations in dairy cows are classified as low (≤140 pg/mL), intermediate (>140 to ≤450 pg/mL), or high (>450 pg/mL). In our study, serum AMH dropped from intermediate to low levels as the summer season progressed. A significant decrease in follicular growth, ovulation rate, and embryo quality was observed in cows averaging serum AMH below 400 pg/mL [22]. In another study, cows with high and intermediate plasma AMH concentrations were 1.42 and 1.51 times more likely to be associated with pregnancy than cows with low plasma AMH concentrations within 84 days following the start of the mating season [49].

High temperatures and humidity in cattle affect ovarian hormone concentrations (i.e., AMH, FSH, LH, etc.), uterine environment, and follicular development, which compromises oocyte competence and early embryonic development [50,51,52]. The fertilization process and embryonic development were disrupted by heat stress, leading to a 30–35% decrease in conception rates in dairy cows [53]. Previous research also demonstrated that heat stress, or hyperthermia, inhibits the growth of ovarian follicles and oocyte competence, leading to a reduction in AMH secretion [22]. Heat stress increases the expression of BCL2L1, a regulator of apoptosis during mammalian ovarian development, among other mechanisms that cause ovarian cells to undergo apoptosis [54]. 

After executing GWAS, we detected 20 SNPs associated with serum AMH that were significant at the Bonferroni adjusted *p*-value. Only 10 of these SNPs were selected as candidate markers because they were located within a gene. Of these SNPs, four of them surpassed the quality control criteria and were tested in a genotype-to-phenotype association study. Such SNPs were rs876084180, rs445674221, rs42849475, and rs8193046, which were within the genes *AMH*, *IGFBP1*, *LRG5*, and *TLR4*. These SNP were then validated as molecular markers for fertility traits and thermotolerance in Holstein cows subjected to a reproductive management program during the summer. 

Nawaz et al. [41] reported candidate genome-wide SNPs on BTA11 and BTA20 associated with AMH concentrations and located in the genes *DENND1A*, *NR5A1*, *NR6A1*, *PTGS1*, *NDUFA8*, and *FST*, which have plausible roles on fertility and AMH regulation. The *DENND1A* gene was also described as a candidate gene associated with a total number of collected and viable embryos in superovulated Canadian Holstein cows [55]. Gobikrushanth et al. [23] reported genome-wide SNPs associated with variation in serum AMH in Holstein cows, one within the *AMH* gene (BTA7) and four in the genes *SCAI* and *PPP6C* (BTA11), and *FGF18* and *EEF2K* (BTA20). A similar GWAS study for AMH concentration in cows from several dairy breeds detected a single quantitative traits locus (QTL) on BTA11. This QTL included candidate genes linked to fertility-related phenotypes, some of them previously reported (*NR5A1*, *HSPA5*, *CRB2*, *DENND1A*, and *NDUFA8*) [49].

The four genome-wide SNPs validated in the current study as markers associated with fertility traits and thermotolerance in Holstein cows were located in the genes *AMH*, *IGFBP1*, *LRG5*, and *TLR4*. These gene loci are on different chromosomes (BTA7, BTA4, BTA5, and BTA8, respectively), which lowers the probability that the validated SNPs are correlated with each other. The SNP rs876084180 was located within the *AMH* (Anti-Müllerian hormone) gene. AMH is a dimeric glycoprotein synthesized by granulosa cells within growing preantral and antral follicles [56]. This protein is considered a potential marker of the ovarian follicular reserve and healthy follicular population in cows [42,46] because it is highly variable, repeatable, and heritable [46,47,48].

In our study, the results suggested that the SNP rs876084180 in the gene *AMH* was a predictor for reproductive traits (i.e., FOL, CL, and CR) because AMH regulates follicular growth and oocyte competence, as well as fertilization and embryo development. Such reproductive functions of AMH appear to be due to its close relationship with the FSH signaling pathway, which regulates and promotes ovarian follicle development [57]. In addition, the ability of AMH to predict RT is because of its direct role of this protein within the TGFβ pathway, which is involved in the architectural remodeling of cells exposed to heat shock [58].

The SNP rs445674221 is within the gene *IGFBP1*. This gene encodes a protein primarily produced in hepatocytes and renal cells, which is a key modulator of IGF1. The IGFBP1 is released into the bloodstream by the liver and binds to IGF1 more strongly than its receptor IGFR1 to control the bioavailability and activity of this hormone [40]. The IGF1 peptide is essential in the regulation of reproductive functions by acting synergistically with gonadotropins [59]. This peptide has an important role in oocyte development, follicular growth, oocyte maturation, ovulation, and luteal function, and also prevents cellular apoptosis and enhanced proliferation in the ovary [60,61,62].

Under HS, IGFBP1 transports IGF1 to the membrane of the oocyte, allowing the interaction with its receptor (IGF1R), which induces activation of the PI3K/AKT signal pathway [63]. Then, AKT inactivates caspases (CASP-3 and CASP-9) [64] and regulates the levels of the BCL-2 family proteins, which reduce cellular alterations caused by HS (i.e., an increase in ROS and H_2_O_2_) [65]. As per activation of the receptor by the IGF1 ligand, the functionality of the mitochondrial membrane is stabilized, the continuation of apoptosis in the oocyte is deferred, and the cell survival pathways are activated [66,67]. Addition of IGF1 (12.5 to 25 ng/mL) to oocytes cultured at 41 °C tended to minimize the reduction in both mitochondrial activity and developmental competence, two processes affected by heat stress [68,69].

The SNP rs42849475 is within the gen *LGR5*. This gene is expressed in the epithelial surface of ovarian stem cells [70], as well as on the surface of the corpus luteum [71]. This cellular location could explain the statistical association of an SNP in the *LGR5* gene with the diameter of FOL and CL in the current study. Similarly, this gene is also expressed in the uterine epithelia, where it appears to collaborate with uterine functions needed to support implantation and maternal recognition of pregnancy [72]; therefore, it seems to explain the associative relationship between the *LGR5* gene and CR.

The *LGR5* gene is a potential target of the Wnt signaling pathway [73], which promotes cell proliferation and cell survival during HS. The proliferative mechanism has been involved in the growth of the mammary gland in dairy cows [74]. This pathway is also associated with energy metabolism, a critical function required to maintain a thermal balance once an animal is exposed to heat stress [75]. Because of this evidence, it is logical to state that there are effects of the *LGR5* gene on thermotolerance-related traits.

The SNP rs8193046 is in the gene *TLR4*. This gene is a crucial mediator of the inflammation-like response in the uterus, triggering the production of regulatory T (Treg) cells, which are required to facilitate and promote embryo implantation [76]. Treg cells suppress anti-fetal effector responses and support the process of uterine vascular remodeling, facilitating strong placental development that supports the growth and survival of the fetus [77,78]. The results of the current study support evidence of the mechanisms through which *TLR4* is helping to improve CR.

As part of the immune response against pathogens and external stressors, *TLR4* induces the expression of pro-inflammatory cytokines [79], which are regulated by nuclear factor kappa-B (NF-κB) expression [80,81]. NF-κB is an important intracellular signaling protein that has a critical role in preventing heat stress-induced early apoptosis and instead promotes growth and survival in cells exposed to heat-stressed conditions [82]. *TLR4* has been associated with a reduction in both lipolytic action and adipocyte mobilization in Holstein cows exposed to heat stress as a body response to minimize heat load [83,84]. It appeared to explain the associative relationship between the *TLR4* gene and thermotolerance traits observed in the current study.

The allele substitution effects detected in the current study supported the concept of the favorable contribution of the genes *AMH*, *IGFBP1*, *LGR5*, and *TLR4* to the reproductive and physiological traits evaluated. This positive effect of the four validated genes was confirmed by the improvement in serum AMH, FOL diameter, CL diameter, and conception rate as the number of SNPs as favorable genotypes increased from 0 to 4. A previous report also involved GWAS, and a validation study identified three genes (i.e., *TLR4*, *GRM8*, and SMAD3) as favorable markers for milk yield and thermotolerance in Holstein cows managed in an HS environment [25]. 

The identification of genetic markers associated with variations in circulating AMH concentrations in Holstein cows exposed to HS would be potentially helpful to identify and select future elite genetic merit cows with greater potential to respond successfully in summer reproductive programs [23]. The discovery and validation of novel molecular markers associated with serum AMH appeared to be a promising strategy to identify dairy cows with superior fertility during summer. Selection of these cows within a genetic improvement program could lead to a faster genetic gain in reproductive efficiency in dairy production systems within semiarid regions [41,42]. As stated by Hansen et al. [85], selecting genetic thermotolerance makes it possible to reduce the impact of HS on reproduction. Genomic studies in Australian Holstein cows reported higher breeding values for fertility in cows that were genetically more thermotolerant [86]. 

A limitation of the current study is the low number of cows included in the GWAS analysis, which, combined with the multiple-testing statistical adjustment, may reduce the number of significant SNPs detected [87]. In addition, due to the polygenic effect of AMH limited variation explained by the markers was expected. Fertility is a highly polygenic trait that is influenced by thousands of SNP effects [23,88], and the candidate genes we detected appeared to be a small number of the many that slightly contribute to the phenotypic variance of serum AMH concentrations. Interestingly, our validation study in an independent Holstein population confirmed the ability of these genes and their variants as predictors for fertility and thermotolerance traits. Because the likelihood of an SNP being significant in two different populations is low, validating the effects of SNPs in independent animal populations seems to be a reliable approach for determining the statistical significance of candidate SNPs [89,90].

## 5. Conclusions

In the current study, we provided evidence that serum AMH concentration was an endocrine biomarker for summer reproductive response, but only when HS did not exceed a moderate threshold. Defining a serum AMH threshold and developing a test executable on a dairy farm would enable farmers to selectively breed cows with elevated AMH levels in order to increase the reproductive efficiency in their herds. GWAS revealed four SNPs associated with serum AMH concentrations, although they only explained a small proportion of variance. A further marker validation study detected the association of these four SNPs with fertility and physiological traits. However, due to the polygenic regulation of the serum AMH concentrations, it is recommended to perform studies using more dense SNP genotyping strategies. Another and possibly better strategy would be to estimate genomic breeding values, which potentially include these types of candidate SNPs. 

## Figures and Tables

**Figure 1 animals-14-00213-f001:**
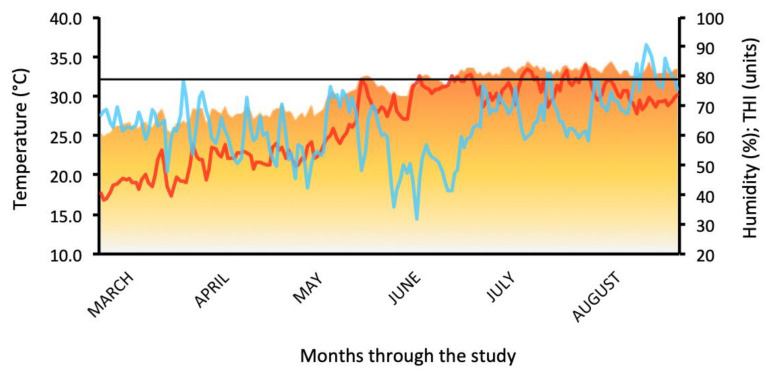
Ambient temperature (AT, °C; red line), relative humidity (RH, %; blue line), and temperature–humidity index (THI, units; orange line) observed during the study. The straight black line indicates the threshold between moderate and severe heat stress. Spring began on 21 March, whereas the summer began on 21 June.

**Figure 2 animals-14-00213-f002:**
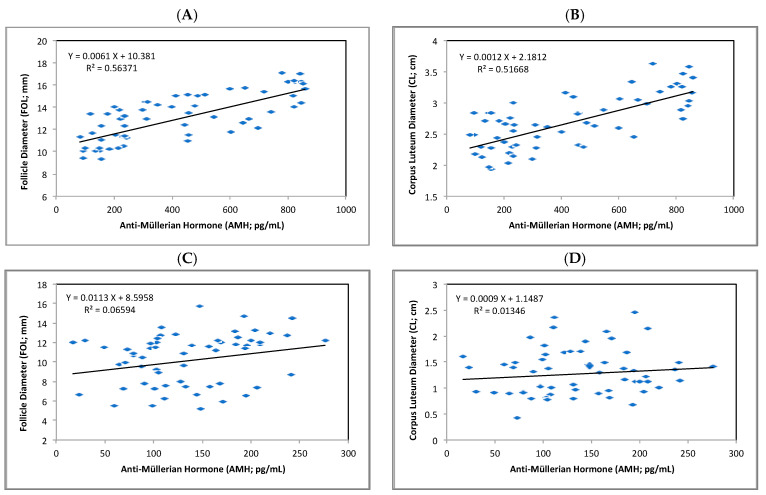
Regression relationship between serum Anti-Müllerian hormone (AMH) and ovarian variables according to the heat-stressed groups. (**A**) FOL diameter vs. serum AMH in MH group, (**B**) CL diameter vs. serum AMH in MH group, (**C**) FOL diameter vs. serum AMH in SH group, and (**D**) CL diameter vs. serum AMH in SH group. In the regression equation, X represents serum AMH concentration, and Y corresponds to FOL or CL diameter.

**Figure 3 animals-14-00213-f003:**
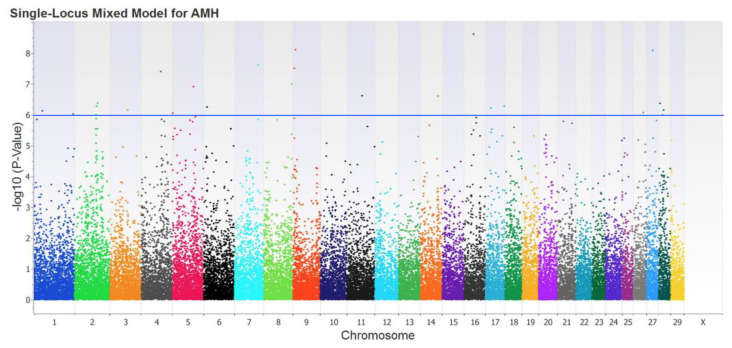
Manhattan plot from single-marker GWAS for Anti-Mülleriam hormone (AMH) serum concentrations in Holstein cows managed during summer in a semi-desert region. The blue line corresponds to a 5% genome-wide threshold (i.e., Bonferroni adjustment).

**Figure 4 animals-14-00213-f004:**
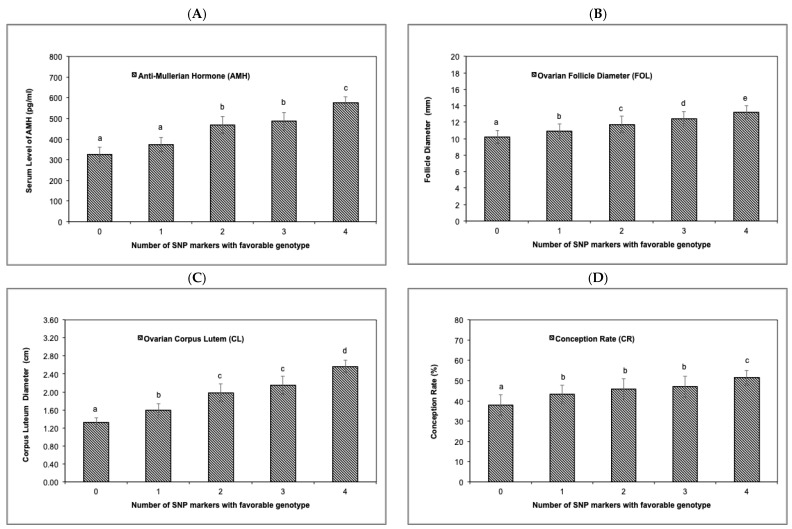
Average values for serum AMH concentrations and reproductive traits (± SE) during the experimental period in Holstein cows according to the number of favorable genotype SNP markers (i.e., 0, 1, 2, 3, or 4). (**A**) Serum AMH concentrations (pg/mL); (**B**) follicle diameter (mm); (**C**) Corpus Luteum diameter (cm); (**D**) conception rate (%). Different letters indicate a significant statistical difference (*p* < 0.05).

**Table 1 animals-14-00213-t001:** Average values ± SE for serum AMH, reproductive traits, and physiological variables in Holstein cows according to heat-stressed environmental conditions.

Variable	Moderate Heat Stress (*n* = 152; THI = 72–79)	Severe Heat Stress (*n* = 128; THI > 79)
AMH (pg/mL)	417.26 ± 33.51 ^a^	136.94 ± 7.65 ^b^
FOL (mm)	12.95 ± 0.93 ^a^	10.15 ± 1.02 ^b^
CL (cm)	2.68 ± 0.12 ^a^	1.26 ± 0.14 ^b^
CR (%)	52.63 ^a^	37.50 ^b^
RT (°C)	38.21 ± 1.25 ^a^	38.73 ± 3.05 ^b^
RR (breaths/min)	63.35 ± 3.14 ^a^	67.14 ± 4.96 ^b^
HT (°C)	35.74 ± 1.12 ^a^	35.60 ± 1.24 ^a^
ST (°C)	36.22 ± 2.18 ^a^	36.31 ± 2.27 ^a^
PT (°C)	36.18 ± 1.64 ^a^	36.23 ± 1.72 ^a^
LT (°C)	35.63 ± 1.86 ^a^	35.68 ± 1.95 ^a^
UT (°C)	36.57 ± 2.12 ^a^	35.59 ± 2.75 ^a^

AMH = serum Anti-Müllerian hormone; FOL = follicle diameter; CL = Corpus Luteum diameter; CR = conception rate; RT = rectal temperature; RR = respiratory rate; HT = head temperature; ST = shoulder temperature; PT = rump temperature; LT = leg temperature; UT = udder temperature. ^a,b^ Different literals between columns indicate a significant statistical difference (*p* < 0.05).

**Table 2 animals-14-00213-t002:** Pearson correlations between serum Anti-Müllerian hormone (AMH) with ovarian and physiological traits in Holstein cows according to climatic experimental groups (HS = heat stress).

Variable	Moderate HS (*p*-Value) 21 May–21 July	Severe HS (*p*-Value) 21 July–21 September
FOL (mm)	0.7508 (<0.0001)	0.2567 (0.0837)
CL (cm)	0.7188 (<0.0001)	0.1160 (0.2155)
RT (°C)	0.3427 (0.0142)	0.0958 (0.2671)
RR (breaths/min)	0.2135 (0.1938)	0.0634 (0.5728)

FOL = follicle diameter; CL = Corpus Luteum diameter; RT = rectal temperature; RR = respiratory rate.

**Table 3 animals-14-00213-t003:** Significant SNPs (*p* < 1.09 × 10^−6^) from a single-marker genome-wide association study (GWAS) with Anti-Müllerian hormone (AMH) in Holstein cows managed in a warm environment.

SNP ID ^1^	Variant ^2^	BTA ^3^	Position ^4^	Gene ^5^	Alleles ^6^	Var ^7^	*p*-Value ^8^
rs41807005	Intergenic	16	34′845050	--------	T/C	1.62	2.35 × 10^−9^
rs109740021	Intergenic	9	9′329501	--------	A/G	1.57	7.53 × 10^−9^
rs42416336	Intronic	27	23′354986	*LONRF1*	A/C	1.52	8.16 × 10^−9^
rs876084180	Intronic	7	21′401999	*AMH*	A/C	1.43	2.38 × 10^−8^
rs136263395	Intergenic	9	4′423733	--------	T/C	1.41	3.11 × 10^−8^
rs445674221	Intronic	4	76′133069	*IGFBP1*	A/G	1.39	3.85 × 10^−8^
rs8193046	Intronic	8	108′833985	*TLR4*	T/C	1.31	9.86 × 10^−8^
rs42849475	Intronic	5	1′087211	*LGR5*	A/G	1.29	1.20 × 10^−7^
rs42338999	Intergenic	11	58′684365	--------	T/C	1.24	2.38 × 10^−7^
rs135450328	Intergenic	14	68′794329	--------	A/C	1.23	2.46 × 10^−7^
rs478504266	Intronic	2	89′091847	*SGO2*	G/A	1.19	4.15 × 10^−7^
rs137194049	Intronic	28	4′632166	*DISC1*	T/C	1.18	4.24 × 10^−7^
rs135441773	Intronic	2	84′744596	*SLC39A10*	T/G	1.17	5.22 × 10^−7^
rs110893810	Intronic	17	73′804897	*RSPH14*	T/C	1.16	5.24 × 10^−7^
rs136745124	Intergenic	6	12′650752	--------	A/G	1.15	5.49 × 10^−7^
rs29024666	Intergenic	17	20′625888	--------	A/C	1.14	6.07 × 10^−7^
rs109442396	Intergenic	28	18′568681	--------	A/C	1.06	6.89 × 10^−7^
rs43475092	Intergenic	3	68′078177	--------	T/C	1.02	7.04 × 10^−7^
rs136711260	Intronic	1	33′715199	CADM2	T/C	0.94	7.31 × 10^−7^
rs133069238	Intergenic	26	38′626896	--------	T/C	0.85	8.33 × 10^−7^

^1^ SNP reference of the NCBI; ^2^ SNP chromosome variant; ^3^ *Bos taurus* autosomal chromosome number; ^4^ SNP position within the chromosome; ^5^ candidate gene symbol (*LONRF1* = LON peptidase N-terminal domain and ring finger 1; *AMH* = Anti-mullerian hormone; *IGFBP1* = insulin-like growth factor binding protein 1; *TLR4* = Toll-like receptor 4; LGR5 = Leucine-rich repeat containing G protein-coupled receptor 5; *SGO2* = Shugoshin 2; *DISC1* = DISC1 scaffold protein; *SLC39A10* = Solute carrier family 39 member 10; *RSPH14* = Radial spoke head 14 homolog; *CADM2* = Cell adhesion molecule 2). ^6^ alleles from the SNP; ^7^ percentage of trait variance explained by the SNP; ^8^ SNP statistical significance.

**Table 4 animals-14-00213-t004:** Identification, gene name, favorable SNP allele, allele frequencies, and Hardy–Weinberg equilibrium analysis for genomic SNPs associated with serum AMH concentrations.

SNP ID ^1^	Gene ^2^	F. Allele ^3^	Allele Frequency ^4^		HWE Test ^5^	HWE *p-*Value ^6^
			A	C		
rs876084180	*AMH*	A	0.35	0.65	0.32	0.46
			A	G		
rs445674221	*IGFBP1*	G	0.53	0.47	2.63	0.18
rs42849475	*LGR5*	G	0.24	0.76	1.25	0.31
rs8193046	*TLR4*	A	0.39	0.61	0.86	0.39

^1^ SNP reference of the NCBI; ^2^ Gene symbol name; ^3^ F. Allele = allele with the favorable effect on phenotype; ^4^ Frequency of both alleles within cow population; ^5^ Hardy–Weinberg equilibrium “χ^2^” test value; ^6^ “χ^2^” test *p*-value with 1 degree of freedom and α = 0.05.

**Table 5 animals-14-00213-t005:** Least-square means ± SE according to SNP’s marker genotypes for serum AMH and reproductive traits in Holstein cattle from a validation population.

SNP ID ^1^	Trait ^2^	Least-Square Means by Genotype ± SE ^3^			*p*-Value ^4^	*AlleleSE* ^5^	*AdditiveFE* ^6^
		AA	AC	CC			
rs876084180	AMH	501.26 ± 45.3 ^a^	382.49 ± 37.9 ^b^	152.94 ± 19.2 ^c^	<0.0001	165.12 *	174.16 *
	FOL	13.01 ± 1.06 ^a^	11.95 ± 1.04 ^b^	10.17 ± 0.98 ^c^	0.0010	1.37 *	1.42 *
	CL	2.53 ± 0.17 ^a^	1.92 ± 0.16 ^ab^	1.47 ± 0.19 ^b^	0.0086	0.50 *	0.53 *
	CR	45.20 ± 3.67 ^a^	36.75 ± 3.32 ^a^	40.65 ± 4.01 ^a^	0.6519	2.21	2.27
		AA	AG	GG			
rs445674221	AMH	141.80 ± 19.6 ^a^	267.39 ± 23.2 ^a^	442.65 ± 37.8 ^b^	0.0009	146.01 *	150.42 *
	FOL	10.63 ± 0.96 ^a^	11.04 ± 1.05 ^a^	13.91 ± 1.12 ^b^	0.0068	1.60 *	1.64 *
	CL	1.24 ± 0.09 ^a^	1.82 ± 0.07 ^b^	2.89 ± 1.18 ^c^	0.0087	0.72 *	0.77 *
	CR	34.50 ± 0.47 ^a^	45.15 ± 0.46 ^b^	56.25 ± 0.42 ^c^	<0.0001	10.55 *	10.87 *
rs42849475	AMH	291.55 ± 26.7 ^a^	342.16 ± 37.5 ^a^	401.29 ± 39.2 ^a^	0.1506	52.76 *	54.87 *
	FOL	9.86 ± 0.87 ^a^	11.16 ± 1.09 ^b^	12.64 ± 1.26 ^b^	0.0079	1.34 *	1.39 *
	CL	1.38 ± 0.07 ^a^	1.92 ± 1.05 ^b^	2.24 ± 1.90 ^b^	0.0096	0.38 *	0.43 *
	CR	39.70 ± 3.07 ^a^	46.20 ± 3.59 ^b^	50.25 ± 4.36 ^b^	0.0188	4.91 *	5.27 *
rs8193046	AMH	474.67 ± 38.7 ^a^	379.23 ± 37.3 ^b^	158.34 ± 14.2 ^c^	<0.0001	151.78 *	158.16 *
	FOL	13.28 ± 1.29 ^a^	11.37 ± 0.94 ^b^	10.14 ± 0.94 ^c^	0.0065	1.52 *	1.57 *
	CL	2.71 ± 2.02 ^a^	1.86 ± 1.45 ^b^	1.20 ± 1.08 ^c^	0.0012	0.67 *	0.75 *
	CR	54.15 ± 3.67 ^a^	45.70 ± 3.29 ^a^	36.75 ± 3.18 ^b^	0.0034	8.45 *	8.70 *

^1^ SNP reference of the NCBI; ^2^ phenotypic traits (AMH = serum Anti-Mullerian hormone, pg/mL; FOL = follicular diameter, mm; CL = Corpus Luteum diameter, cm; CR = conception rate, %); ^3^ least-square means according to SNP genotype ± SE (^a,b,c^ indicate statistical difference among genotypes at); ^4^ *p*-value = statistical significance; ^5^ Allele substitution effect; ^6^ Additive fixed estimated effect; * indicate significance at *p* < 0.05.

**Table 6 animals-14-00213-t006:** Least-square means ± SE according to SNP’s marker genotypes for physiological traits in Holstein cattle from a validation population.

SNP ID ^1^	Trait ^2^	Least-Square Means by Genotype ± SE ^3^			*p*-Value ^4^	*AlleleSE* ^5^	*AdditiveFE* ^6^
		AA	AC	CC			
rs876084180	RT	38.14 ± 2.98 ^a^	38.25 ± 2.67 ^a^	38.79 ± 3.39 ^b^	0.0316	−0.28 *	0.32 *
	RR	63.67 ± 5.22 ^a^	65.44 ± 5.19 ^a^	66.35 ± 5.19 ^a^	0.0928	−1.63 *	1.69 *
		AA	AG	GG			
rs445674221	RT	38.89 ± 3.02 ^a^	38.46 ± 2.96 ^b^	38.15 ± 2.77 ^c^	0.0094	−0.34 *	0.37 *
	RR	68.32 ± 4.98 ^a^	65.01 ± 4.67 ^b^	61.96 ± 4.12 ^c^	0.0067	−3.06 *	3.18 *
rs42849475	RT	38.81 ± 3.54 ^a^	38.39 ± 2.69 ^b^	38.26 ± 3.82 ^b^	0.0235	−0.24 *	0.27 *
	RR	65.79 ± 4.28 ^a^	64.03 ± 5.02 ^a^	62.99 ± 4.67 ^a^	0.2138	−1.35 *	1.40 *
rs8193046	RT	38.16 ± 2.65 ^a^	38.49 ± 3.08 ^b^	38.93 ± 3.19 ^c^	0.0076	−0.35 *	0.38 *
	RR	61.24 ± 5.09 ^a^	64.96 ± 5.16 ^b^	68.95 ± 4.87 ^c^	0.0052	−3.69 *	3.85 *

^1^ SNP reference of the NCBI; ^2^ phenotypic traits (RT = rectal temperature, °C; RR = respiration rate, breaths/min); ^3^ least-square means according to SNP genotype ± SE (^a,b,c^ indicate statistical difference among genotypes); ^4^ *p*-value = statistical significance; ^5^ Allele substitution effect; ^6^ Additive fixed estimated effect; * indicate significance at *p* < 0.05.

## Data Availability

The data that support the findings of this study are available from the corresponding author, P.L.-N., upon reasonable request. The productive data are not publicly available due to they belong to the records of the cooperating dairy farmers.

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
