# Peer review of "The Anti-Müllerian Hormone as Endocrine and Molecular Marker Associated with Reproductive Performance in Holstein Dairy Cows Exposed to Heat Stress"

_animals, 2024, doi:10.3390/ani14020213_

Round 1

Reviewer 1 Report

Comments and Suggestions for Authors

The research presents “evaluate associations of serum AMH concentrations and AMH-related single nucleotide polymorphisms (SNPs) with reproductive performance in Holstein dairy cows exposed to heat-stressed environmental conditions”. This research provides interesting information. However, some important changes need to be made before final publication.

Abstract: review the guidelines of the “Journal”. The following is mentioned in “MDPI Style Guide”: “The abstract contains a summary of the entire paper and can be up to 200 words long with only one paragraph.” (https://www.mdpi.com/authors/layout).

MATERIAL AND METHODS

General comments: I recommend being more specific about how the 300 cows were distributed since in the results in table 1, 280 animals are reported and the remaining 20?

Specific comments:

Line 116.- mention “Three hundred lactating and spring-calved Holstein cows with average body weight of 635.1 ± 32.5 kg, body condition score from 2.5 to 3.5, and 4 to 6 yrs-old were used in this study.” I recommend being more specific regarding the selection criteria, such as lactations, days in milk, etc.

Line 199-200.- they mention “During the course of the study, both rectal temperature (RT; °C) and respiratory rate 199 (RR; breaths/min) were measured at 1200 h two times per week”. Who makes these measurements. Regarding the temperature, why did they not consider that they only measured at 1200h? And why didn't they take more measurements from other areas?

Line 205-207.- they mention “Data on the ambient temperature (AT; °C) and relative humidity (RH; %) were obtained from the REMAS website (http://www.siafeson.com/remas) by a local meteorological station (Network of Automatic Meteorological Stations of Sonora)”. Why didn't they take direct data from the dairy herd? I recommend mentioning the distance between the “local meteorological station” and the place where the study was carried out. Did the corrals have tools to reduce the heat?

RESULTS

Line 258.- Figure 1, I recommend indicating when “summer” begins

Line 278.- mention “Pearson correlations between serum Anti-Müllerian hormone (AMH) with ovarian and physiological traits in Holstein cows according to climatic experimental groups (HS = heat stress)”. Was the data normally distributed?

Line 288.- Figure 2 restructure this figure. Homogenize the format and also the legends have errors (A, B, C and D).

DISCUSSION

Line 405-406.- they mention “However, this predictive ability was not detected in Holstein cows when HS progressed from moderate to severe during summer”. What could have been the reason?

Line 421-423.- mention “Reductions in hematocrit and modifications in water balance during heat stress may attribute to the decline in hormone concentrations within peripheral blood [42]”. However, you did not determine hormonal levels, this may be speculative since you do not have a reference range. It would have been interesting for at least a subgroup to have been able to determine them (FSH and P4), to be able to relate it.

Author Response

We thank the Animals editorial office and the Reviewers for their comments and suggestions that helped to improve the quality of the manuscript. We have provided response to each comment and suggestion made by the Reviewers. Revisions made to the manuscript were highlighted in yellow.

Review Report (Reviewer 2):

The research presents “evaluate associations of serum AMH concentrations and AMH-related single nucleotide polymorphisms (SNPs) with reproductive performance in Holstein dairy cows exposed to heat-stressed environmental conditions”. This research provides interesting information. However, some important changes need to be made before final publication.

Abstract: review the guidelines of the “Journal”. The following is mentioned in “MDPI Style Guide”: “The abstract contains a summary of the entire paper and can be up to 200 words long with only one paragraph.” (https://www.mdpi.com/authors/layout).

Response: We have reviewed and adjusted the abstract to a maximum of 200 words in length.

MATERIAL AND METHODS

General comments: I recommend being more specific about how the 300 cows were distributed since in the results in table 1, 280 animals are reported and the remaining 20?

Response: Thank you for your suggestion. We clarified in Materials and Methods section that we used an initial Holstein cattle population (n=300) to measure serum AMH and perform a genomic study (GWAS). Later, we used an additional Holstein cow population (n=280) from other dairy herds to perform the SNP marker validation study.

Specific comments:

Line 116.- mention “Three hundred lactating and spring-calved Holstein cows with average body weight of 635.1 ± 32.5 kg, body condition score from 2.5 to 3.5, and 4 to 6 yrs-old were used in this study.” I recommend being more specific regarding the selection criteria, such as lactations, days in milk, etc.

Response: We have added the suggested information.

Line 199-200.- they mention “During the course of the study, both rectal temperature (RT; °C) and respiratory rate 199 (RR; breaths/min) were measured at 1200 h two times per week”. Who makes these measurements.

Response: We have clarified that the same person (i.e., a trained graduate student) collected all physiological measures through the study.

Regarding the temperature, why did they not consider that they only measured at 1200h?

Response: I would like to clarify that we collected rectal temperature at 07:00 and 17:00 hours, in the holding pen of the milking parlor, just before morning and afternoon milking. I apologize for this involuntary mistake, which has been corrected in the manuscript.

And why didn't they take more measurements from other areas?

Response: We also collected hair coat temperatures (°C) from the head, shoulder, rump, legs and udder. They were measured with an infrared thermometer (model 568 Fluke®, Fluke Corp., WA, USA; accuracy ± 1°C) positioned approximately 3 m of distance from the animal. These measures did not differ between MH and SH climatic groups, then we decided to focus only in rectal temperature as a more direct measure of the physiological body temperature. We have added this information in the manuscript.

Line 205-207.- they mention “Data on the ambient temperature (AT; °C) and relative humidity (RH; %) were obtained from the REMAS website (http://www.siafeson.com/remas) by a local meteorological station (Network of Automatic Meteorological Stations of Sonora)”. Why didn't they take direct data from the dairy herd?

Response: We also used environmental sensors (i.e., Kestrel DROP data loggers) to collect ambient data inside the dairy herd, which were highly correlated with data from REMAS website. However, we decided to use data collected from the meterological station because REMAS is the official website that we have reported in our previous articles. Then, we just tried to be consistent with the source of the meterological data.

I recommend mentioning the distance between the “local meteorological station” and the place where the study was carried out.

Response: We have added the suggested information.

Did the corrals have tools to reduce the heat?

Response: In the corrals there were no tools to reduce heat stress. The cows received cooling with sprayers and fans located in the holding pen of the milking parlor two times per day.

RESULTS

Line 258.- Figure 1, I recommend indicating when “summer” begins

Response: We have added the suggested information.

Line 278.- mention “Pearson correlations between serum Anti-Müllerian hormone (AMH) with ovarian and physiological traits in Holstein cows according to climatic experimental groups (HS = heat stress)”. Was the data normally distributed?

Response: Yes, all data collected from the variables serum AMH, FOL diameter and CL diameter were normally distributed. We have added this information in the manuscript.

Line 288.- Figure 2 restructure this figure. Homogenize the format and also the legends have errors (A, B, C and D).

Response: We have made these corrections.

DISCUSSION

Line 405-406.- they mention “However, this predictive ability was not detected in Holstein cows when HS progressed from moderate to severe during summer”. What could have been the reason?

Response: We assumed that Holstein cows are able to maintain physiological functions when they are exposed to moderate heat stress. However, severe heat stress appeared to affect the ability of the follicle to synthesize AMH leading to a potential imbalance between follicle size and AMH secretion. We have added this information in the manuscript.

Line 421-423.- mention “Reductions in hematocrit and modifications in water balance during heat stress may attribute to the decline in hormone concentrations within peripheral blood [42]”. However, you did not determine hormonal levels, this may be speculative since you do not have a reference range. It would have been interesting for at least a subgroup to have been able to determine them (FSH and P4), to be able to relate it.

Response: We have removed this paragraph and included information more specific about heat stress effects on AMH secretion.

Reviewer 2 Report

Comments and Suggestions for Authors

In the present study, serum AMH and AMH-associated SNPs were investigated on their use as predictors for reproductive performance in heat-stressed dairy cows.The GWAS included 300 Holstein cows with serum AMH measurements. A validation study consisted of two groups including cows under moderate (n=152) and severe heat-stress (n=128). Serum AMH seemed useful as predictor of fertility measures in cows under moderate heat stress. Serum AMH concentrations and reproductive traits appeared asssociated with four intragenic SNPs.

Comments

Simple abstract:

Lines 13-18: please focus on AMH, fertility and heat stress. This part is too general for fertilty. The main focus is on reproduction under heat stress in dairy cows.

Line 19-20: please outline this issue and say which problems are encountered here.

Lines 22-24: is there a contradiction with serum AMH levels. How to use these results remains unclear. Please say how farmers could benefit from your results.

Line 32: in which influence these SNPs the AMH gene. Association was with serum AMH levels. Do you mean regulation of AMH levels or do you think there are structural variants of proteins affecting transport, processing or metabolism of AMH.

Line 39-41: Possible benefits for farmers should be emphasized.

Keywords are not appropriate as these should not be part of the title.

Introduction

There some important references missing.

A supplementary Table summarizing the previous results would be useful for heritabilities, GWAS-results and candidate genes infleunceing AMH levels.

Heat stress has to be defined and explained in miore detail.

Lines 45-58: very general. Please focus on heat stress and which parameters appear useful for investigating reproduction under heat stress.

Lines 66-71: "Anti-Müllerian hormone (AMH) has emerged as potential biomarker" please say it more concrete such as serum or plasma levels, log-transformed and when measured (diurnal secretion, time in estrus cycle, season). Are there reference values for breeds and THI?

Lines 73-81: were these issues not yet answered by previous reports. Please re-write and focus on the open questions.

Line 115: how THI was calculated.

Line 125: do not understand "early in the summer", please say at which time of the day, cycle stage and in which period.

Line 155: please explain the fixed effects applied in the model. The SNP effect is missing in the model (fixed or random model?) 

Line 156: AMH level as measured or log-transformed: please explain and justify whether observed levels were normally distributed.

Line 225-244: validation study: did you include the heat stress group in the model? SNP-genotype: was this done SNP-by-SNP or all SNPs simultaneously.

Were the SNPs correlated. Did you test for multicollinearity of SNPs tested herein.

What do you mean with "lactation"?

Line 266-290: which model was applied for the analysis.

Line 236: the covariant: linear covariate? Please clarify.

Why no animal model for the validation study?

Line 248: THI is not explained.

Figure 1: days of the year: not clear, please clarify. please indicate season.

Table 1: SE or STD?

Figure 2: equations in Figure 2 are not readable. Please improve.

Table 3: Var: are the proportions shown really percentages. This means that less than 1% of the phenotypic variance is captured by a single SNP.

Table 4: F. Allele: there seems something wrong such as rs876084180, AMH with alleles A and C and favourable allele T. Please check the contents of this Table.

Line 326: are the LS-means from the model estimates.

Tables 5 and 6: please check superscripts, there are typos.

Line 349: 4 p-value = statistical significance; 5 Allele substitution effect; 4 Additive fixed estimated effect: confusing. The same on lines 353-354.

Lines 355-373: combined effects in methods not described. Collinearity checked between SNPs?

Did you test for heat stress group differences?

Conclusions are to general and focussed on the outcomes of the present study. Please revise. Also add the benefits for the farmers or how to translate your results into practical work on farms or whether additional research is needed before application on farms.

Comments on the Quality of English Language

No comments

Author Response

We thank the Animals editorial office and the Reviewers for their comments and suggestions that helped to improve the quality of the manuscript. We have provided response to each comment and suggestion made by the Reviewers. Revisions made to the manuscript were highlighted in yellow.

Review Report (Reviewer 3):

In the present study, serum AMH and AMH-associated SNPs were investigated on their use as predictors for reproductive performance in heat-stressed dairy cows.The GWAS included 300 Holstein cows with serum AMH measurements. A validation study consisted of two groups including cows under moderate (n=152) and severe heat-stress (n=128). Serum AMH seemed useful as predictor of fertility measures in cows under moderate heat stress. Serum AMH concentrations and reproductive traits appeared associated with four intragenic SNPs.

Comments

Simple abstract:

Lines 13-18: please focus on AMH, fertility and heat stress. This part is too general for fertilty. The main focus is on reproduction under heat stress in dairy cows.

Response: Edited as suggested.

Line 19-20: please outline this issue and say which problems are encountered here.

Response: Edited as suggested.

Lines 22-24: is there a contradiction with serum AMH levels. How to use these results remains unclear. Please say how farmers could benefit from your results.

Response: We have clarified that AMH as endocrine marker appeared to be effective in cows affected by moderate heat stress but not in cows under severe heat stress. However, AMH as genetic marker appeared to be effective in cows exposed to either moderate or severe heat stress. We have also added benefits of these results to the farmers.

Line 32: in which influence these SNPs the AMH gene. Association was with serum AMH levels. Do you mean regulation of AMH levels or do you think there are structural variants of proteins affecting transport, processing or metabolism of AMH.

Response: Thank you for your interesting comment. We think that the four significants SNPs are gene variants influencing pathways that regulate AMH levels. Perhaps these SNP mutations occur in regulatory regions of the genes AMH, IGFBP1, LGR5 and TLR4, which are associated with the variability in circulating concentrations of AMH. We have added this information starting the Discussion section (i.e., ending the first paragraph). We also provided more details of the involved pathways in the second part of the Discussion section.

Line 39-41: Possible benefits for farmers should be emphasized.

Response: We have added this information at the end of the Abstract.

Keywords are not appropriate as these should not be part of the title.

Response: Edited as suggested.

Introduction

There some important references missing.

A supplementary Table summarizing the previous results would be useful for heritabilities, GWAS-results and candidate genes influenceing AMH levels.

Response: A supplementary table has been added as suggested.

Heat stress has to be defined and explained in more detail.

Response: Edited as suggested.

Lines 45-58: very general. Please focus on heat stress and which parameters appear useful for investigating reproduction under heat stress.

Response: We have replaced general paragraphs by information that is focused on heat stress and parameters to study fertility and heat stress.

Lines 66-71: "Anti-Müllerian hormone (AMH) has emerged as potential biomarker" please say it more concrete such as serum or plasma levels, log-transformed and when measured (diurnal secretion, time in estrus cycle, season).

Response: We have added the suggested information.

Are there reference values for breeds and THI?

Response: Reference values for AMH concentration differed between Bos indicus (770 pg/ml) and Bos taurus (330 pg/ml) cattle breeds. Serum concentration of AMH also differed between cool (1018.8 pg/ml) and warm (321.4 pg/ml) seasons in beef cattle. However, AMH concentration is reported very stable among age groups and stages of the estrus cycle in Holstein cows. This information has been added in the manuscript.

Lines 73-81: were these issues not yet answered by previous reports. Please re-write and focus on the open questions.

Response: We have rewritten the paragraph and focused on the main topic of this study.

Line 115: how THI was calculated.

Response: The THI was calculated using the equation proposed by Mader et al. (2010): THI = (0.8 AT) + [(RH/100) (AT 14.4)] + 46.4. However, we have rewritten the paragraph because THI calculation is described later in the section “2.8 Physiological traits and climatic data”.

Line 125: do not understand "early in the summer", please say at which time of the day, cycle stage and in which period.

Response: Edited as suggested.

Line 155: please explain the fixed effects applied in the model.

Response: We have made these correction.

The SNP effect is missing in the model (fixed or random model?)

Response: The SNP effect has been included in the model. This model included fixed and random effects.

Line 156: AMH level as measured or log-transformed: please explain and justify whether observed levels were normally distributed.

Response: Serum AMH concentrations were first log-transformed to meet the assumptions for a Gaussian distribution. For ease of interpretation, the transformed least squares means results were back transformed. Log-transformed AMH values demonstrating a far better approximation to normality, then AMH concentrations were normally distributed.

Line 225-244: validation study: did you include the heat stress group in the model?

Response: Yes, I included the heat stress group in the model but I forgot to report it in the manuscript. I apologize for this unintentional omission, which has been corrected.

SNP-genotype: was this done SNP-by-SNP or all SNPs simultaneously.

Response: Genotyping was performed analyzing all SNPs simultaneously.

Were the SNPs correlated. Did you test for multicollinearity of SNPs tested herein.

Response: We consider that testing for multi-collinearity is a very good strategy to detect associations between SNPs ; however, for SNP data, typically, it’s the Linkage Disequilibrium (LD) that is first estimated. If there is no LD, then can fit all the SNP in the model; however, if there is LD, and a “tag” SNP is determined, then can run the model with only Tag SNP. Can then test multi-collinearity of the Tag SNP. In the current study we did not detect LD or linear correlation between the significant SNPs, which suggests that it is not necessary to perform the multi-collinearity test. We have added this information in section “2.5 Multiple testing adjustment and linkage disequilibirum (LD) analysis”.

What do you mean with "lactation"?

Response: We have replaced the term “lactation” by “parity”. Lactation refers to the number of completed lactation periods from each cow.

Line 266-290: which model was applied for the analysis.

Response: A simple linear regression analysis was executed in PROC REG including serum AMH as predictor for FOL and CL diameters. The model or linear regression equation was as follows: Yi = β0 + β1 X, where Yi is the response or dependent variable (i.e., FOL or CL diameters), β0 is the intercept coefficient, β1 is the coefficient of X, and X is the predictor or independent variable (i.e., serum AMH concentration). This information has been added in the section “2.10. Statistics for the validation analysis”.

Line 236: the covariant: linear covariate? Please clarify.

Response: We have clarified that it was a linear covariate.

Why no animal model for the validation study?

Response: We have added the associative statistical model in section “2.10. Statistics for the validation analysis”. This is the model that our research group has used and reported previously to validate SNPs as molecular markers..

Line 248: THI is not explained.

Response: We have added an explanation for THI.

Figure 1: days of the year: not clear, please clarify. please indicate season.

Response: Edited as suggested.

Table 1: SE or STD?

Response: It has been clarified in Table 1 (i.e., average values ± SE).

Figure 2: equations in Figure 2 are not readable. Please improve.

Response: Edited as suggested.

Table 3: Var: are the proportions shown really percentages. This means that less than 1% of the phenotypic variance is captured by a single SNP.

Response: We have corrected these values in Table 3. I apologize for the involuntary mistake.

Table 4: F. Allele: there seems something wrong such as rs876084180, AMH with alleles A and C and favourable allele T. Please check the contents of this Table.

Response: Thank you for your comment. We have corrected these values for “F. Allele” column in Table 4.

Line 326: are the LS-means from the model estimates.

Response: Yes, these are the LS-means of the estimates from the associative statistical model.

Tables 5 and 6: please check superscripts, there are typos.

Response: We have made these corrections.

Line 349: 4 p-value = statistical significance; 5 Allele substitution effect; 4 Additive fixed estimated effect: confusing. The same on lines 353-354.

Response: Edited as suggested to clarify the information.

Lines 355-373: combined effects in methods not described.

Response: We have added this missing information at the end of Methods section.

Collinearity checked between SNPs?

Response: We performed Linkage Disequilibrium (LD) analysis and the R2 value confirmed that there were no associations between the SNPs. Such results suggested that it is not necessary to further perform the collinearity test.

Did you test for heat stress group differences?

Response: Yes, we have added this information in the first paragraph of the section “2.10. Statistics for the validation analysis”.

Conclusions are too general and focussed on the outcomes of the present study. Please revise. Also add the benefits for the farmers or how to translate your results into practical work on farms or whether additional research is needed before application on farms.

Response: We have revised and rewritten Conclusions incorporating your valuable suggestions.

Round 2

Reviewer 2 Report

Comments and Suggestions for Authors

The authors improved their manuscript at many places.

However, some comments and recommendations are still necessary.

Line 187: all SNP were fitted simultaneously: this seems not to be correct. 

This would be an overfitting with more than 50,000 SNPs as fixed effects. There was a misunderstanding in my last report where I asked if the analysis is done SNP by SNP. I did not ask for the lab analysis because you used beadchips for genotyping but for the statistical method. Usually GWAS are dealing each SNP separately from the other SNPs. Otherwise, you have to apply multivariate methods and these are working quite different.

Line 376 "single-locus mixed model" and Legend of Figure 3: you are saying "single marker GWAS" . This means that you did the GWAS SNP by SNP. Please correct the mistake in the Methods section.

Please add the Bonferroni significance threshold for p < 0.05 in Figure 3.

Figure 2 has to be improved. The regression equation is misprinted.

Line 198: not clear where to find these results. On line 385 you give genomic heritability. Did you really calculate this estimate using ASREML? Please clarify and add the ASREML estimate.

Line 201: € please change to small epsilon.

Line 278: Finally, genotyping and analysis of PCR data were performed using StepOne software (Life Technologies Corporation, version 2.3).

>>> may this correct: Finally, genotype calling and analysis of PCR data were performed using StepOne software (Life Technologies Corporation, version 2.3).

Genotyping cannot be done using software.

Line 641: Molecular analysis  >> GWAS

Line 644: These SNPs are proposed as markers for genetic programs to improve fertility in Holstein cows exposed to a heat-stressed. : please remove this sentence becaus there is a contradiction in your conclusions: You are saying the trait (AMH serum levels) is polygenic, and therefore, markers are useless as they explain only a very small part of the variance (as shown in your manuscript).

Another and possibly better strategy would be to employ genomic breeding values which capture the genomic variation much better than a set of SNPs.

You should consider this in your conclusions.

Comments on the Quality of English Language

Minor edits

Author Response

We thank the Animals Assistant Editor and the Reviewers for comments and suggestions that helped to improve the quality of the manuscript. We have provided response to each comment and suggestion made by the Reviewers. Revisions made to the manuscript were highlighted in yellow.

I would like to kindly let you all know that we put forth substantial effort to improve the English and grammar of the manuscript.

Review Report (Reviewer 3):

The authors improved their manuscript at many places.

However, some comments and recommendations are still necessary.

Line 187: all SNP were fitted simultaneously: this seems not to be correct.

This would be an overfitting with more than 50,000 SNPs as fixed effects. There was a misunderstanding in my last report where I asked if the analysis is done SNP by SNP. I did not ask for the lab analysis because you used beadchips for genotyping but for the statistical method. Usually GWAS are dealing each SNP separately from the other SNPs. Otherwise, you have to apply multivariate methods and these are working quite different.

Line 376 "single-locus mixed model" and Legend of Figure 3: you are saying "single marker GWAS" . This means that you did the GWAS SNP by SNP. Please correct the mistake in the Methods section.

Response: Thank you for your comment, we have edited as suggested.

Please add the Bonferroni significance threshold for p < 0.05 in Figure 3.

Response: Edited as suggested.

Figure 2 has to be improved. The regression equation is misprinted.

Response: Edited as suggested.

Line 198: not clear where to find these results. On line 385 you give genomic heritability. Did you really calculate this estimate using ASREML? Please clarify and add the ASREML estimate.

Response: Thank you for your comments. I would like to kindly clarify that we assessed the variance explained by the SNPs associated with serum AMH concentrations using the SNP Variation Suite software (SVS, Version 8.8.1), through the EMMA algorithm for genomic REML. However, I was confused by the term “genomic REML” and a term described in the SVS output as “pseudo-heritability”, but we did not really use ASREML and we did not calculate the ASREML estimate for heritability. I apologize for this misunderstanding. Therefore, we have removed the information about genomic heritability calculation and genomic heritability result from the manuscript.

Line 201: € please change to small epsilon.

Response: Edited as suggested.

Line 278: Finally, genotyping and analysis of PCR data were performed using StepOne software (Life Technologies Corporation, version 2.3).

>>> may this correct: Finally, genotype calling and analysis of PCR data were performed using StepOne software (Life Technologies Corporation, version 2.3).

Genotyping cannot be done using software.

Response: Thank you for your comment. We have edited as suggested.

Line 641: Molecular analysis >> GWAS

Response: Edited as suggested.

Line 644: These SNPs are proposed as markers for genetic programs to improve fertility in Holstein cows exposed to a heat-stressed. : please remove this sentence becaus there is a contradiction in your conclusions: You are saying the trait (AMH serum levels) is polygenic, and therefore, markers are useless as they explain only a very small part of the variance (as shown in your manuscript).

Another and possibly better strategy would be to employ genomic breeding values which capture the genomic variation much better than a set of SNPs.

You should consider this in your conclusions.

Response: Thank you for your interesting comments, they were considered in our conclusions as suggested.